# The Combination of Upconversion Nanoparticles and Perovskite Quantum Dots with Temperature-Dependent Emission Colors for Dual-Mode Anti-Counterfeiting Applications

**DOI:** 10.3390/nano13243102

**Published:** 2023-12-08

**Authors:** Qun Zhang, Yuefeng Gao, Lihong Cheng, You Li, Sai Xu, Baojiu Chen

**Affiliations:** 1School of Science, Dalian Maritime University, Dalian 116026, China; qunmay529@sina.com (Q.Z.); lyzhx0519@163.com (Y.L.); bjchen@dlmu.edu.cn (B.C.); 2Marine Engineering College, Dalian Maritime University, Dalian 116026, China

**Keywords:** anti-counterfeiting, upconversion nanoparticles, perovskite quantum dots, thermal enhancement luminescence

## Abstract

Novel and high-security anti-counterfeiting technology has always been the focus of attention and research. This work proposes a nanocomposite combination of upconversion nanoparticles (UCNPs) and perovskite quantum dots (PeQDs) to achieve color-adjustable dual-mode luminescence anti-counterfeiting. Firstly, a series of NaGdF_4_: Yb/Tm UCNPs with different sizes were synthesized, and their thermal-enhanced upconversion luminescence performances were investigated. The upconversion luminescence (UCL) intensity of the samples increases with rising temperature, and the UCL thermal enhancement factor rises as the particle size decreases. This intriguing thermal enhancement phenomenon can be attributed to the mitigation of surface luminescence quenching. Furthermore, CsPbBr_3_ PeQDs were well adhered to the surfaces and surroundings of the UCNPs. Leveraging energy transfer and the contrasting temperature responses of UCNPs and PeQDs, this nanocomposite was utilized as a dual-mode thermochromic anti-counterfeiting system. As the temperature increases, the color of the composite changes from green to pink under 980 nm excitation, while it displays green to non-luminescence under 365 nm excitation. This new anti-counterfeiting material, with its high security and convenience, has great potential in anti-counterfeiting applications.

## 1. Introduction

Anti-counterfeiting technology has been in the spotlight in recent years due to the rapid development of technology. Counterfeit products have become a particularly serious global problem due to the increasing prevalence of counterfeiters. The legitimate rights and interests of businesses and consumers have been violated [1]. Thus far, watermarking technology [2], barcode technology [3], laser holography [4], and fluorescent anti-counterfeiting technology [5] have been developed to increase the difficulty of detecting counterfeit products and authenticating genuine ones. Among these techniques, fluorescence anti-counterfeiting techniques have received a lot of attention due to their easy implementation, inapplicability for replication, low cost, and high concealment properties [6]. Currently, the fluorescent materials used in the anti-counterfeiting fields are mainly carbon-based quantum dots [7], perovskite quantum dots [8], conjugated polymers [9], rare-earth luminescent materials [10], etc.

UCNPs have peculiar anti-Stokes luminescence and narrow band emission, long fluorescence lifetime, and superior optical and chemical stability [11,12,13]. Moreover, the nature of UCNPs near-infrared light excitation can effectively reduce background fluorescence. However, conventional single-mode luminescence anti-counterfeiting materials and single-detection light sources still cannot solve the problem of forgery [14]. Therefore, it is essential to further develop UCNP-based anti-counterfeiting materials with more flexible color changes to achieve high-level security anti-counterfeiting. Normally, the realization of anti-counterfeiting based on UCNPs mainly depends on their diverse luminescence properties, such as excitation wavelength-dependent luminescence [15], temperature-dependent luminescence [16], excitation power-dependent luminescence [17], time-resolved luminescence [18], etc. In general, due to the temperature-quenching effect of UCNPs, the luminescence intensity decreases at high temperatures. In recent years, many research groups have found abnormal thermal enhanced upconversion luminescence contrary to traditional thermal quenching. Wang et al. [19] found size-dependent anomalous luminescence thermal enhancement behavior in NaGdF_4_@NaGdF_4_:Yb^3+^/Tm^3+^ inert-core/active-shell UCNPs; they attributed this enhancement to the alleviation of surface quenching caused by lattice thermal expansion. However, Jin’s group [20] believed that thermally favorable phonon on the nanoparticle surface provided a critical energy mismatch between sensitizer and activator, thereby activating the dark surface layer. In addition, some research groups [16,21,22,23] have proposed that the thermal enhancement phenomenon is caused by the release of water molecules on the surface of the nanoparticles. These results open up new possibilities for UCNPs as thermal-induced fluorescent anti-counterfeiting materials. CsPbBr_3_ PeQDs have excellent optical properties, including large absorption coefficients, high photoluminescence quantum yields (PLQYs), and tunable emission wavelengths, which are also required for good fluorescent anti-counterfeiting materials [24]. Moreover, the fluorescence emission intensity of CsPbBr_3_ PeQDs is also strongly affected by temperature [25]. The combination of UCNPs and CsPbBr_3_ PeQDs may result in more flexible color variations depending on temperature. CsPbBr_3_ exhibits bright visible light excited by UV or blue light. UCNPs can convert near-infrared photons into high-energy ultraviolet or visible light under 980 nm laser excitation [26]. So, the near-infrared sensitizer UCNPs can be used as the energy donors in the energy transfer (ET) process (non-radiative Forster resonance energy transfer (FRET) or radiative photon resorption (PR)) [27], providing an energy transfer path for CsPbBr_3_ PeQDs. Obviously, UCNPs combined with CsPbBr_3_ can achieve the dual-mode luminescence of the two components. Moreover, the distinct temperature-dependent luminescence behaviors of UCNPs and CsPbBr_3_ PeQDs could potentially result in temperature-responsive emissions, leading to tunable changes in color emissions.

For this article, we first synthesized a series of NaGdF_4_: Yb/Tm UNCPs with different sizes and investigated their thermal enhanced upconversion luminescence performances. It is shown that the UCL intensity of the samples all increased with increasing temperature and that the UCL thermal enhancement factor increased with decreasing size. The anomalous thermal enhancement phenomenon can be attributed to the alleviation of surface luminescence quenching. Meanwhile, it has been proved that CsPbBr_3_ PeQDs can be well adhered to the surfaces and surroundings of UCNPs [27]. Therefore, the nanocomposite exhibits dual-mode luminescence properties under 980 nm and 365 nm excitation. Based on the energy transfer and the opposite temperature response of UCNPs to PeQDs, the nanocomposite was used as a dual-mode thermochromic anti-counterfeiting system.

## 2. Experimental Section

### 2.1. Materials

1-Octadecene (ODE, C_18_H_36_, >90.0%, Aladdin, Shanghai, China), Oleic acid (OA, C_18_H_34_O_2_, AR, Aladdin, Shanghai, China), YbCl_3_·6H_2_O (99.9%, Aladdin, Shanghai, China), GdCl_3_·6H_2_O (99.9%, Aladdin, Shanghai, China), TmCl_3_·6H_2_O(99.9%, Aladdin, Shanghai, China), sodium hydroxide (NaOH, AR, Tianjin Jinbei Fine chemical Co., LID, Tianjin, China), ammonium fluoride (NH_4_F,96%, Tianjin Guangfu Technoligy development Co., LID, Tianjin, China), methanol(CH_3_OH, AR, Shenyang Xinxing Reagents Factory, Shenyang, China), Cs_2_CO_3_ (99.9%, Aladdin), PbBr_2_ (99.0%, Aladdin, Shanghai, China), and oleylamine (OLA, 80–90%, Aladdin, Shanghai, China) were used as received without further purification.

### 2.2. The Synthesis of the NaGdF_4_: Yb/Tm Core UCNPs

Briefly, 1 mmol RECl_3_·6H_2_O (RE: 0.798 mmol Gd^3+^, 0.2 mmol Yb^3+^, and 0.002 mmol Tm^3+^) was mixed in a 50 mL three-necked flask and 6 mL OA and 15 mL ODE, and a solid–liquid mixture was obtained. Then, the solid–liquid mixture was heated to 150 °C under a N_2_ atmosphere until the solid was completely dissolved in liquid. Next, the mixed liquid was cooled to 30 °C. Subsequently, 0.1 gNaOH and 0.148 gNH_4_F were dissolved in 5 mL methanol solution, and methanol solution was slowly added into the mixed liquid in the flask and held for 30 min. Under a N_2_ atmosphere, the mixed liquid was heated to 100 °C for 1 h to evaporate methanol and water before being heated to 270–310 °C (for 1 h). Afterwards, the mixed liquid cooled naturally to room temperature. Furthermore, 20 mL ethanol was added into the cooled mixed liquid. Finally, the products were collected by centrifugation, washed several times with cyclohexane–ethanol, and dispersed in 1 mL cyclohexane.

### 2.3. The Synthesis of the NaGdF_4_: Yb/Tm@NaGdF_4_: Yb Core–Shell UCNPs

Briefly, 1 mmol RECl_3_·6H_2_O (RE: 0.798 mmol Gd^3+^, 0.2 mmol Yb^3+^, and 0.002 mmol Tm^3+^) was mixed in a 50 mL three-necked flask and 6 mL OA and 15 mL ODE, and a solid–liquid mixture was obtained. Then, the solid–liquid mixture was heated to 150 °C under a N_2_ atmosphere until the solid was completely dissolved in liquid. Next, the mixed liquid was cooled to 30 °C. The core UCNPs (prepared under 300 °C) dispersed in cyclohexane were slowly dropped in to the mixed liquid for ten minutes. Subsequently, 0.1 gNaOH and 0.148 gNH_4_F were dissolved in 5 mL methanol solution, and methanol solution was slowly added into the mixed liquid in the flask and held for 30 min. Under a N_2_ atmosphere, the mixed liquid was heated to 100 °C for 1 h to evaporate methanol and water before being heated to 270–310 °C (for 1 h). Afterwards, the mixed liquid cooled naturally to room temperature. Furthermore, 20 mL ethanol was added into the cooled mixed liquid. Finally, the products were collected by centrifugation, washed several times with cyclohexane–ethanol, and dispersed in 1 mL cyclohexane.

### 2.4. The Preparation of the Cs-Oleate

Cs_2_CO_3_ (0.8140 g) was loaded into a 50 mL 3-neck flask along with ODE (30 mL) and OA (2.5 mL); the mixture was dried for 1 h at 120 °C and then heated under N_2_ to 150 °C until the solution was clear. Since Cs-oleate precipitates out of ODE at room temperature, it has to be preheated to 100 °C before use.

### 2.5. The Synthesis of the UCNP-CsPbBr_3_ Nanocomposites

ODE (10 mL) and 0.376 mmol PbBr_2_ were added to a 50 mL three-necked flask. Then, the solution was heated to 120 °C and kept in the N_2_ atmosphere for 40 min. Subsequently, 1 mL OLA and 0.6 mL OA and core/shell UCNPs (1 mmol) dispersed in cyclohexane (1 mL) were added into the mixture, and the cyclohexane was evaporated at 120 °C. The solution was heated up to 160 °C after PbBr_2_ was entirely dissolved; then, Cs-oleate (0.8 mL) was quickly added. After 20 s, it was cooled via an ice water bath and washed 3 times with cyclohexane before being vacuum-dried for 8 h.

### 2.6. Characterization

The samples dispersed in cyclohexane were dropped onto the copper net, and their surface morphologies were characterized using a JEM-2000EX 120 kV transmission electron microscope (TEM). The energy-dispersive X-ray (EDX) images of the samples dispersed in cyclohexane and dropped onto the copper net were characterized using FEI Tecnai G2 F20, and a spectral model was constructed using OXFORD X-max 80 T. The crystallization characteristics of the sample powder were measured using a Bruker D8 Advance powder X-ray diffractometer at a scanning rate of 5°min^−1^ in a 2θ range (from 10° to 60°) using Cu Kα radiation (λ = 1.5418 Å) operating at a voltage of 40 kV and a current that was maintained at 40 mA. The temperature-independent UC PL spectra of the sample powders were measured under excitation using an external 980 nm fiber laser using a Hitachi F-4600 fluorescence spectrometer with a self-made temperature controller. The UV/visible absorption spectra of the CsPbBr_3_ dispersed in cyclohexane were determined using a Shimadzu UV-3600PC with a UV-visible scanning spectrophotometer operating in the range of 200 to 800 nm. The fluorescence decay curves of the sample powders were measured using a time-dependent single photon counting (TCSPC) technique on the FS5 fluorescence spectrometer. Multi-color images were taken under NIR irradiation by using a Xiaomi phone with a UV-IR-cutoff filter.

## 3. Results and Discussion

### 3.1. Morphologies and Structures of the UCNPs and UCNP-CsPbBr_3_ Nanocomposites

The preparation of UCNP–CsPbBr_3_ (UCQ) nanocomposites mainly consisted of the following three parts: Firstly, co-doped (Yb^3+^ and Tm^3+^) NaGdF_4_ UCNPs were prepared using the high-temperature solvothermal method described in the Experimental Section. Then, a NaGdF_4_: Yb shell was coated using the same method to enhance the UCL strength of the UNCPs. Subsequently, in the presence of OLA and Cs- OA precursors, OA-capped UCNPs were used to guide the attachment of CsPbBr_3_ from PbBr_2_. Subsequently, by using two different ligands (OA of UCNPs CsCO_3_ and OLA of PbBr_2_), CsPbBr_3_ PeQDs were well adhered to the surfaces and surroundings of the UCNPs. Finally, a UCQ nanocomposite, combining the advantages of UCNPs and PeQDs, was formed. To investigate the size dependency of the thermo-enhanced luminescence [23], the NaGdF_4_: Yb/Tm UCNPs were prepared under different temperatures (270 °C, 280 °C, 300 °C, 310 °C). The samples of the UCNPs with different synthesis temperatures dispersed in cyclohexane and dropped onto the copper mesh were characterized by transmission electron microscopy (TEM). TEM images of the UCNPs are shown in Figure 1a–d. It can be seen that all the UCNPs are single-dispersed and homogeneous. The average sizes of the UCNPs increased with an increase in the reaction temperature, and they were determined to be 4.94 ± 0.08 nm, 5.43 ± 0.11 nm, 18.98 ± 0.02 nm, 66.09 ± 0.14 nm, respectively, for the samples prepared at 270 °C, 280 °C, 300 °C, and 310 °C, as shown in Appendix A. Figure 1h shows the XRD patterns of the as-prepared UCNPs and the standard cards of the cubic-phase NaGdF_4_ (PDF#27-0699) and hexagonal-phase NaGdF_4_ (PDF #27-0697) as controls [28]. As can be seen, the products prepared at 270 °C are the mixture of the two phases, and when the temperature increases to 280 °C and above, the UCNPs completely transfer to the hexagonal phase. In addition, NaGdF_4_: Yb core–shell UCNPs were also prepared under 300 °C to investigate the effect of the active shell on the thermo-enhanced luminescence. As shown in Figure 1e, the size of the core–shell UCNPs is 39.20 nm, an increase, on average, by 10.11 nm compared with that of the core UCNPs (prepared under 300 °C). This indicates that the active shell was successfully coated on the core UCNPs. A TEM image of the UCQ composite is shown in Figure 1f; CsPbBr_3_ PeQDs with a diameter of 11~13 nm are attached to and around the surfaces of the UCNPs. From the energy-dispersive X-ray spectrometry (EDS) mapping image shown in Figure 1g, it can be seen that Gd, F, Br, Yb, Tm, and Cs are uniformly distributed and covered with each other, which further indicates the successful preparation of the nanocomposites. In addition, the elemental mapping of NaGdF_4_: Yb/Tm UCNPs was also measured, as shown in Appendix A. From Appendix A, it can be seen that the elements of F, Na, Gd, Yb, and Tm are homogeneously distributed in the UCNPs.

### 3.2. Thermo-Enhanced Luminescence of the NaGdF_4_: Yb/Tm UCNPs

A series of NaGdF_4_: Yb/Tm UCNPs with different sizes were prepared to investigate the size-dependence of the thermo-enhanced luminescence. The temperature-dependent UCL emission spectra of the group of UCNPs in the 303–483 K temperature range are shown in Figure 2a–d. Under 980 nm excitation, the UCNPs exhibit the typical intrinsic emission lines of Tm^3+^, namely 450 nm, 473 nm, 645 nm, 695 nm, and 803 nm, which originate from the transitions of ^1^D_2_→^3^F_4_, ^1^G_4_→^3^H_6_, ^1^G_4_→^3^F_4_, ^3^F_2,3_→^3^H_6,_
^3^H_5_→^3^H_6_ [29]. Interestingly, significant luminescence enhancement was observed in all samples, which is in opposition to the traditional thermal quenching phenomenon of UCL luminescence. By calculating the integral luminescence intensities of the ^1^G_4_→^3^H_6_ transition of Tm^3+^, as shown in the inset of Figure 2a–d (the integral luminescence intensities of ^1^D_2_→^3^F_4_ and ^1^G_4_→^3^F_4_ are shown in Appendix A), it can be seen that the maximum thermal enhancement factors of NaGdF_4_: Yb/Tm are increased by 6.45-, 3.93-, 3.61-, and 1.20-fold from 303 to 483 K for the nanoparticle sizes of 4.94 nm, 5.43 nm, 18.98 nm, and 66.09 nm, respectively. It is worth noting that the ^3^H_5_→^3^H_6_ transition of Tm^3+^ also shows the same thermal enhancement trend. As shown in Appendix A, the maximum thermal enhancement factor of the UCNPs synthesized at 300 °C is 1.51. Obviously, the UCL thermal enhancement ratio decreases with the increase in size. Here, we speculate that the enhanced thermal induction of UCL may be related to the quenching effect of surface-absorbed moisture. The Hydroxyl vibrations on the surfaces of the UCNPs cause the multi-phonon deactivation of the Yb^3+^ sensitizers [16]. The release of moisture molecules from the surfaces of the UCNPs at high temperatures leads to a decrease in the nonradiative relaxation rate of Yb^3+^ [16]. Although the OA ligand on the surface of the nanoparticle protects Yb^3+^, at room temperature, small water molecules in the environment can still diffuse to the nanoparticle surface, accelerating the non-radiative relaxation rate of Yb^3+^ [30]. The surface/volume ratios of small-size UCNPs are higher than those of larger ones; their surface quenching plays an important role in determining their luminescence intensity [31]. Therefore, small-size UCNPs adsorb more water molecules, resulting in more obvious thermal enhancement on them. It is worth mentioning that the temperature-dependent UCL emission spectra displayed by all samples were immediately measured when the corresponding temperature wass reached, and the water molecules adsorbed on their surface may not have been completely removed. Therefore, larger samples exhibit more obvious thermal quenching when the temperature continues to increase. To verify this, we measured the temperature-dependence UCL spectra of active shell- and inert shell-coated UCNPs. As shown in Figure 2e,f, the integrated intensities of the ^1^G_4_→^3^H_6_ transition increased by 2.10- and 1.30-fold for the active shell- and inert shell-coated UCNPs, respectively. The inert shell can largely shield Yb^3+^ from the surface quenching [32] of water molecules, leading to a smaller UCL enhancement factor at high temperatures. Moreover, constant-temperature UCL spectra of UCNPs at 333 K, 363 K, and 393 K were obtained. As shown in Figure 3a, when the ambient temperature around the samples was maintained at 333 K, the UCL intensity increased significantly in the first 20 min and slowed down over 40 min. When the temperature increased to 363 K, the trend regarding UCL intensity was similar to that at 333 K (In Figure 3b). When the temperature continued to rise to 393 K, no obvious UCL enhancement was observed within 60 min (In Figure 3c). This also confirmed that some water molecules were not removed in the temperature-dependent UCL emission measurement, and the surface hydroxyl vibrations still accelerated the non-radiative relaxation of Yb^3+^. When the water molecules on the surfaces of the UCNPs were completely removed at 333 K and 363 K in the measurement of the constant-temperature UCL spectra, the integrated intensity of Tm^3+^ in the-constant temperature UCL spectrum at 393 K remained almost unchanged. In Figure 3d–f, the integral luminescence intensities of ^1^D_2_→^3^F_4_ and ^1^G_4_→^3^F_4_ are shown; the variation trends are same as that of the ^1^G_4_→^3^H_6_ transition.

### 3.3. Energy Transfer Mechanism and Anti-Counterfeiting Applications of the UCQ Composite

As is known, the luminescence intensity of CsPbBr_3_ PeQDs decreases with increasing temperature due to the thermal quenching effect [33]. Thus, the distinct temperature-dependent luminescence behaviors of UCNPs and CsPbBr_3_ PeQDs hold promise for enabling temperature-responsive emissions, thereby facilitating customizable shifts in color emissions. Here, NaGdF_4_: Yb/Tm@ NaGdF_4_: Yb UCNPs and CsPbBr_3_ PeQDs combine in the presence of OLA and Cs-OA precursors, and OA-capped UCNPs were used to guide CsPbBr_3_ attachment based on the PbBr_2_ method, whereby UCNPs serve as an effective sensitizer for PeQDs to boost their luminescence via FRET [27,34,35,36,37]. An efficient FRET process requires the following essential conditions: (1) the distance between the donor and the acceptor must be close (typically less than 10 nm), and (2) the absorption spectrum of the acceptor overlaps with the emission spectrum of the donor [34]. In this study, CsPbBr_3_ PeQDs were successfully attached to the surfaces of the UCNPs, as shown in Figure 1e, which indicates that the distance between the energy donor and the acceptor was shortened, providing the first condition for the FRET process. Figure 4a shows the UCL spectrum of NaGdF_4_: Yb/Tm nanoparticles under 980 nm excitation (purple line), as well as the absorption (blue line) and emission (green line) spectra of the CsPbBr_3_ PeQDs. The absorption band of the PeQDs is centered at 473 nm, displaying great overlap with the 450 nm and 473 nm emissions of Tm^3+^ ions, providing the second condition for the FRET process. A comparison between the UCL spectra of the nanocomposites and the content of PeQD precursors is shown in Figure 4b. As is shown, the emission intensity of the PeQDs gradually increases with the increase in PeQD precursors, while the blue emission of Tm^3+^ decreases. The red emission of Tm^3+^ at 630–680 nm is almost unchanged. It is worth mentioning that in the presence of the PeQDs, when only non-radiative FRET occurs from Tm^3+^ to the PeQDs, the intensity ratio of the ^1^D_2_→^3^F_4_ transition at 450 nm and the ^1^G_4_→^3^H_6_ transition at 473 nm remains unaltered. This constancy arises from the shared origin of these transitions within the same excited state. Consequently, it is suggested that in the presence of PeQDs, radiative PR takes place from Tm^3+^ to the QDs. Specifically, the emission from the ^1^G_4_→^3^H_6_ transition of Tm^3+^ is reabsorbed by the PeQDs, inducing a discernible alteration in the intensity ratio between the ^1^D_2_→^3^F_4_ emission (450 nm) and the ^1^G_4_→^3^H_6_ emission (473 nm).

To corroborate the FRET process, the luminescence dynamic curves of the ^1^G_4_→^3^H_6_ transition for Tm^3+^ with different concentrations of PeQD precursors were determined. All the dynamic curves exhibited the same damped trends. A comparison between the constants of the decay lifetimes and the content of PeQD precursors is exhibited in Figure 4c. Evidently, as the content of PeQD precursors rises from 0 to 0.752 mmol, the decay lifetime for the ^1^G_4_→^3^H_6_ transition of Tm^3+^ diminishes from 859.5 to 600.9 μs. This observed trend significantly confirms the proposition of the non-radiative FRET occurring from the ^1^G_4_ level of Tm^3+^ towards the PeQDs. The calculation of the FRET efficiency from the ^1^G_4_ level of Tm^3+^ to the PeQDs can be conducted through using the following equation [38]:$$\eta =1-\frac{\tau_{c}}{\tau_{0}}$$
where τ_0_ and τ*_c_* are the decay lifetimes of the ^1^G_4_ level of Tm^3+^ in the absence and presence of PeQDs, respectively. As shown in Figure 4d, through using the above equation, the FRET efficiency is estimated to be 18.4%, 27.1%, and 30.1%, corresponding to precursor contents of 0.188 mmol, 0.564 mmol, and 0.752 mmol, respectively.

A schematic representation of the observed energy transfer upconversion processes is provided in Figure 4e. Under 980 nm irradiation, Yb^3+^ ions are excited from ^2^F_7/2_ to ^2^F_5/2_ due to the large absorption cross-section centered at around 980 nm. Then, the excited electrons are transferred to the higher energy levels of Tm^3+^ (^1^D_2_, ^1^G_4_, ^3^F_3_) and undergo population through multi-photon processes [39]. The various excited states of Tm^3+^ lead to the emission peaks of 450 nm (^1^D_2_→^3^F_4_), 473 nm (^1^G_4_→^3^H_6_), 645 nm (^1^G_4_→^3^F_4_), 695 nm (^3^F_2,3_→^3^H_6_), and 803 nm (^3^H_5_→^3^H_6_). Notably, the energy level spacing of the ^1^D_2_ → ^3^F_4_ and ^1^G_4_ → ^3^H_6_ transitions corresponds to the energy gap between the conduction band (CB) and valence band (VB) of the CsPbBr_3_ PeQDs [40,41]. It is crucial to emphasize that the energy transfer occurs through both non-radiative and radiative mechanisms. Consequently, based on the comprehensive findings of this study, the coexistence of both these mechanisms has been conclusively established.

Based on the energy transfer of UCNPs to PeQDs and the opposite temperature response, a dual-mode thermochromic anti-counterfeiting system has been constructed. The temperature-dependent spectra of the UCQ nanocomposite under 980 nm excitation were obtained and are shown in Figure 5a,b. With the increase in temperature, the luminescence intensity of Tm^3+^ at the energy levels of ^1^D_2_→^3^F_4_ and ^1^G_4_→^3^H_6_ is significantly enhanced. In contrast, the luminescence intensity of CsPbBr_3_ decreases. The UCQ nanocomposite exhibits flexible color variations depending on temperature. Moreover, the nanocomposite shows significant temperature-dependent color bright changes under 365 nm excitation. As shown in Figure 5c, in the range of 303–483 K, under 365 nm excitation, the luminescence intensity of the UCQ nanocomposite gradually decreases, and its spectral shape and luminescence center do not change depending on temperature. In order to evaluate the potential applications of the prepared materials in anti-counterfeiting, herein, a pattern of the moon was prepared by utilizing the UCQ nanocomposite. As shown in Figure 5d, under 980 nm excitation, during the heating process, the color of the pattern significantly changed from green to pink from 303 K to 483 K. However, the color of the pattern excited at 365 nm faded from 303 K to 363 K before disappearing completely.

## 4. Conclusions

In conclusion, our work introduces an innovative nanocomposite, which was created by integrating UCNPs and PeQDs, for the development of a versatile dual-mode luminescence anti-counterfeiting system. Initially, a series of differently sized NaGdF_4_: Yb, Tm UNCPs were synthesized and subjected to an investigation of their thermal-enhanced upconversion luminescence. Notably, the UCL intensity increased proportionally with increasing temperature, while the UCL thermal enhancement factor exhibited a corresponding increase as particle size diminished. This intriguing thermal enhancement effect is ascribed to the alleviation of surface luminescence quenching. Moreover, a nanocomposite was crafted by combining small-sized UCNPs with PeQDs. Capitalizing on energy transfer and the divergent temperature responses of UCNPs and PeQDs, this nanocomposite was harnessed as a dual-mode thermochromic anti-counterfeiting system. As the temperature increases, the nanocomposite shows different color variations under 980 nm and 365 nm excitation. Evidently, this novel anti-counterfeiting material, distinguished by its high security and user-friendly attributes, holds substantial promise for diverse anti-counterfeiting applications.

## Figures and Tables

**Figure 1 nanomaterials-13-03102-f001:**
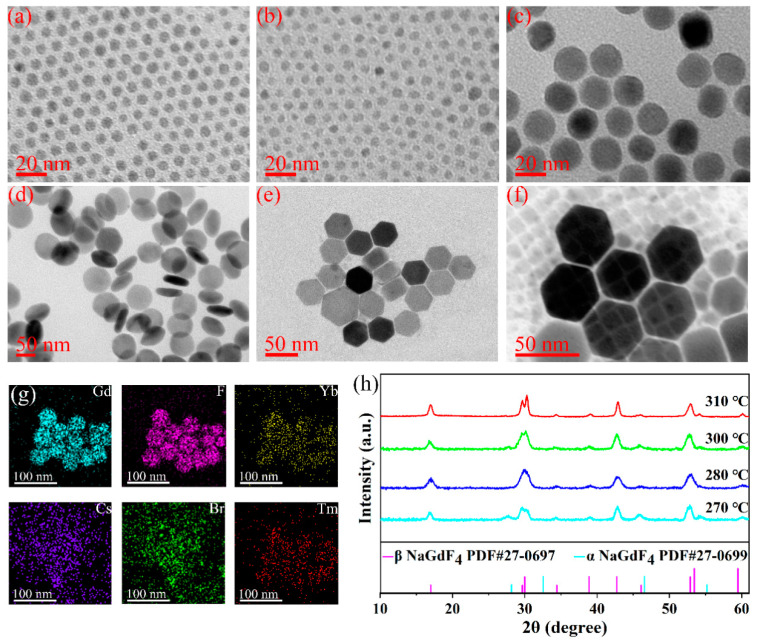
TEM images of NaGdF_4_: Yb/Tm synthesized at (**a**) 270 °C, (**b**) 280 °C, (**c**) 300 °C, (**d**) 310 °C; (**e**) TEM images of core–shell UCNPs synthesized at 300 °C; (**f**) TEM images of the UCQ nanocomposite; (**g**) EDS mapping images of the UCQ nanocomposite; (**h**) XRD patterns of NaGdF_4_: Yb/Tm UCNPs prepared at different temperatures.

**Figure 2 nanomaterials-13-03102-f002:**
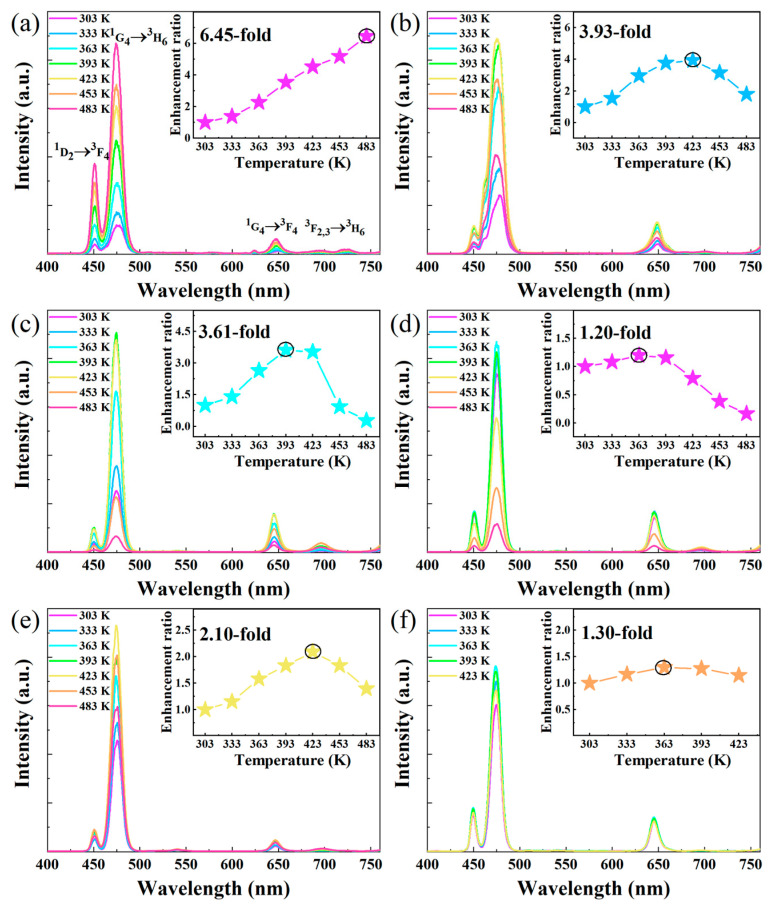
Temperature-dependent UCL emission spectra of NaGdF_4_:Yb/Tm UCNPs with sizes of (**a**) 4.94 nm, (**b**) 5.43 nm, (**c**) 18.98 nm, and (**d**) 66.09 nm. Temperature-dependent UCL emission spectra of (**e**) core @ active shell UCNPs and (**f**) core @ inert shell UCNPs.

**Figure 3 nanomaterials-13-03102-f003:**
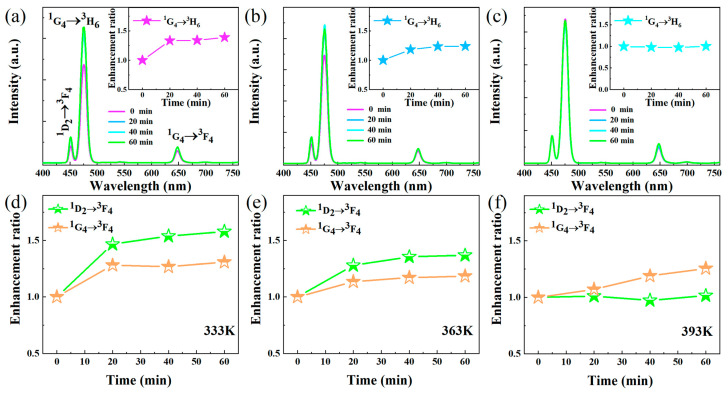
Constant-temperature emission spectra at (**a**) 333 K, (**b**) 363 K, and (**c**) 393 K. ^1^D_2_→^3^F_4_ and ^1^G_4_→^3^F_4_ integral luminescence intensities at (**d**) 333 K, (**e**) 363 K, and (**f**) 393 K.

**Figure 4 nanomaterials-13-03102-f004:**
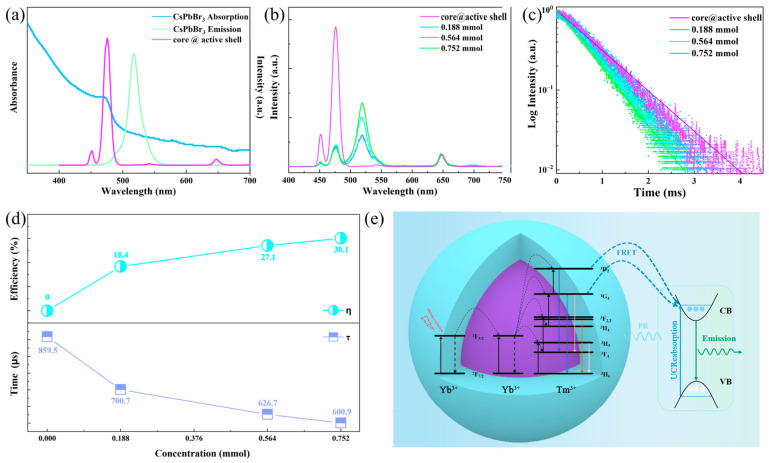
(**a**) Absorption and emission spectra of CsPbBr_3_ PeQDs and emission spectrum of NaGdF_4_: Yb/Tm UCNPs. (**b**) Emission spectra and (**c**) luminescence decay curves of the UCQ composite with different concentrations of PeQDs; (**d**) FRET efficiency and decay lifetime tendency of the UCQ composite with different PeQD precursors; (**e**) energy transfer process from UCNPs to PeQDs.

**Figure 5 nanomaterials-13-03102-f005:**
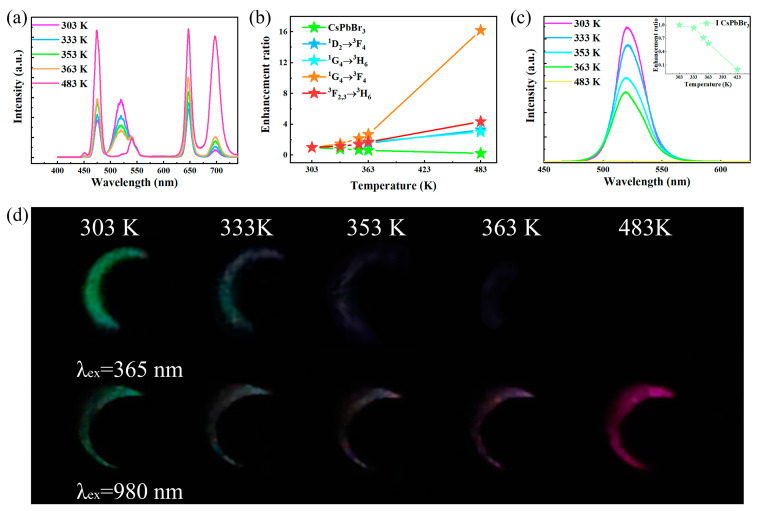
(**a**) Temperature-dependent emission spectra of UCQ under 980 nm excitation; (**b**) integral fluorescence intensity of Tm^3+^ and CsPbBr_3_; (**c**) temperature-dependent emission spectra of UCQ under 365 nm excitation; (**d**) photographs of the UCQ nanocomposite at different temperatures under 365 nm and 980 nm excitation.

## Data Availability

The data presented in this study are available from the corresponding authors upon reasonable request.

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
