# Peer review of "The Combination of Upconversion Nanoparticles and Perovskite Quantum Dots with Temperature-Dependent Emission Colors for Dual-Mode Anti-Counterfeiting Applications"

_nanomaterials, 2023, doi:10.3390/nano13243102_

Round 1

Reviewer 1 Report

Comments and Suggestions for Authors

What is "Cu-KR radiation (1.54178A)" and  Shimazu?

Schematic diagram of the process for preparing UCQ nanocomposite is meaningless, because it is not based on data and balls is an internet simplification of synthesis, and it is not a diagram only images proposed by the authors . It must be removed or presented in a scientific manner.

How  average sizes of UCNPs are determined to be 4.94 nm? and what is the error of determination +/_ ?

Because TEM images of NaGdF4: Yb/Tm synthesized at fig 1c, 1d, 1e are not homogeneous the mapping of UCQ at fig 1g must be extended on Yb3+ and Tm3+. Moreover elemental   mapping of NaGdF4: Yb/Tm UCNPs must be measured as well, because CsPbBr3 PeQDs were not analyzed quantitatively by other analytical methods (ICP - MS).

The speculation "the enhanced thermal induction of UCL may be related to the 202 quenching effect of surface-absorbed moisture. " has no evidence in the spectra of water on the surface, which were not measured.  Moreover it was observed at " temperature higher than 423 K"?

Because it is important feature for UCL study, the IR spectra must be studied otherwise it is a speculation which is not acceptable for publication at the high quality journal.

Comments on the Quality of English Language

What is "Cu-KR radiation (1.54178A)" and  Shimazu?

Reviewer 2 Report

Comments and Suggestions for Authors

The paper by Zhang et al. reports on the synthesis and characterization of a nanocomposite of NaGdF4:Yb,Tm upconversion nanoparticles and CsPbBr3 perovskite quantum dots for use as a thermochromic anti-counterfeiting system. The chosen subject is very topical and the authors reveal an interesting approach of employing energy transfer between the different luminescent species to induce the observed temperature-dependent changes in the emission characteristics. They study the influence of various factors on the photoluminescence of the upconversion nanoparticles and the nanocomposite, including the particle size, temperature, and duration of illumination. Finally, the authors demonstrate an anti-counterfeiting system with different thermal dependencies for various excitation wavelengths. Whilst the paper shows very interesting results worth to be disseminated, it contains some inconsistencies, which should be addressed before being recommended for publication:

-         The paper does not explicitly mention, in which form the samples are prepared for the optical (PL, UV-vis) measurements: in solution or in film.

-         Please carefully fix the typos and grammar throughout the text.

-         In Line 62, “considered to be” could be substituted with “required for”.

-         In Line 105, “After” could be substituted with “Afterwards”.

-         In Line 174, add “by” after “increases”.

-         In Line 180, “successful preparation of the nanocomposites is mentioned”. Whereas the synthesis of the composite is not doubted, it is difficult to unambiguously agree with the authors on the structure of the obtained nanocomposite. Based on the TEM image, it is difficult to confirm that PeQD are covering the surface of UCNPs, as shown in Scheme 1. Moreover, EDS mapping images show that the perovskite phase is distributed across the whole probed region, and not exclusively on the surface of UCNPs. This questions again that the obtained particles have the same end structure, as that shown in Scheme 1. This requires clarification or reconsideration.

-         In Line 188, add “of” before “thermo-enhanced”.

-         In Line 201, add “rises” before “higher”.

-         In Line 228, “shown” should be moved to the end of the sentence as “is shown”.

-         In Line 232, the Caption sentence is not complete.

-         In Line 257, add “concentration” after “precursor”.

-         In Line 285, the 1I63F4 transition is not indicated in the diagram in Fig. 4e.

-         In Lines 293-294, “carried out” could be substituted with “obtained”.

-         In Lines 297-298, it is written “Moreover, the nanocomposite shows significant temperature-dependent color changes under 365 nm excitation.” However, according to Fig. 5c, the shape of the spectrum and thus color do not change with temperature. This must be clarified.

-         In Lines 308-309, it is written: “… (c) luminescence decay curves of UCQ composite with different PeQDs precursors…”. However, the data seems to refer not to different precursors, but to different concentrations of the same precursor.

Round 2

Reviewer 1 Report

Comments and Suggestions for Authors

Response is adequate and manuscript can be published.

Comments on the Quality of English Language

Acceptable.